# Kidney-Specific Membrane-Bound Serine Proteases CAP1/Prss8 and CAP3/St14 Affect ENaC Subunit Abundances but Not Its Activity

**DOI:** 10.3390/cells12192342

**Published:** 2023-09-23

**Authors:** Elodie Ehret, Sévan Stroh, Muriel Auberson, Frédérique Ino, Yannick Jäger, Marc Maillard, Roman Szabo, Thomas H. Bugge, Simona Frateschi, Edith Hummler

**Affiliations:** 1Department of Biomedical Sciences, Faculty of Biology and Medicine, University of Lausanne, 1011 Lausanne, Switzerland; elodie.ehret@unil.ch (E.E.);; 2National Center of Competence in Research “Kidney.CH”, 1011 Lausanne, Switzerland; 3Department of Medicine, University of Geneva, 1211 Geneva, Switzerland; 4Department of Pharmacology, Max-Planck-Institute for Heart and Lung Research, 61231 Bad Nauheim, Germany; 5Service of Nephrology, Department of Medicine, Lausanne University Hospital (CHUV), 1005 Lausanne, Switzerland; 6National Institutes of Health/NIDCR, Bethesda, MD 20892, USA

**Keywords:** proteolytic ENaC activation, CAP1/Prss8, CAP3/St14, sodium and potassium balance

## Abstract

The serine proteases CAP1/Prss8 and CAP3/St14 are identified as ENaC channel-activating proteases in vitro, highly suggesting that they are required for proteolytic activation of ENaC in vivo. The present study tested whether CAP3/St14 is relevant for renal proteolytic ENaC activation and affects ENaC-mediated Na^+^ absorption following Na^+^ deprivation conditions. CAP3/St14 knockout mice exhibit a significant decrease in CAP1/Prss8 protein expression with altered ENaC subunit and decreased pNCC protein abundances but overall maintain sodium balance. RNAscope-based analyses reveal co-expression of CAP3/St14 and CAP1/Prss8 with alpha ENaC in distal tubules of the cortex from wild-type mice. Double CAP1/Prss8; CAP3/St14-deficiency maintained Na^+^ and K^+^ balance on a Na^+^-deprived diet, restored ENaC subunit protein abundances but showed reduced NCC activity under Na^+^ deprivation. Overall, our data clearly show that CAP3/St14 is not required for direct proteolytic activation of ENaC but for its protein abundance. Our study reveals a complex regulation of ENaC by these serine proteases on the expression level rather than on its proteolytic activation.

## 1. Introduction

Hypertension is strongly associated with an increased risk of cardiovascular events, stroke, and kidney disease [1]. Blood pressure depends on salt balance, although the salt-induced blood pressure response varies within a population [2]. Increased Na^+^-transport via the renal epithelial sodium channel (ENaC) was proposed to be a common phenomenon in salt-dependent forms of hypertension [3]. Trypsin-like proteases were first identified as new autocrine regulators of ENaC in vitro [4]. Later, due to their homology, further membrane-bound serine proteases, like CAP1/Prss8 (also termed channel-activating protease 1, prostasin), CAP2/Tmprss4 (channel-activating protease 2) or CAP3/St14 (channel-activating protease 3, matriptase) were found to induce amiloride-sensitive Na^+^ current when co-expressed with ENaC in Xenopus oocytes [5,6]. Additional to these membrane-bound serine proteases, the channel-activating proteases (CAPs) comprise intracellular or soluble serine proteases, the lysosomal cysteine protease cathepsin-S, the metalloproteinase meprin, and the vesicular mannan-binding lectin associated serine protease-1 (MASP-2) (see for review [7]). A sequential cleavage of the extracellular domains, releasing an inhibitory tract within single ENaC subunits, was proposed [8]. An in vivo regulatory role for these proteases in ENaC-mediated Na^+^ absorption has so far been confirmed for CAP1/Prss8 in the lung and colon (see for review [7]), kallikrein in the colon and kidney [9], and urokinase plasminogen activator in the lung [10]. None of these tested proteases turned out to be required for a direct proteolytic ENaC activation and thus an ENaC-mediated Na^+^ balance (see for review [7]). The membrane-bound serine protease CAP2/Tmprss4 was instead involved in renal water handling upon dietary K^+^ depletion [11].

The membrane-bound serine proteases CAP1/Prss8 and CAP3/St14 belong to the S1A subfamily of serine proteases, are produced as inactive zymogens requiring activation by further proteolytic cleavage and are both tightly regulated by HAI-1 and HAI-2 (hepatocyte growth factor activator inhibitor type 1 and type 2) [12]. CAP1/Prss8 was linked to hypertension in rats when overexpressed [13], to a moderate increase of human blood pressure in a genetic variant of CAP1/Prss8 [14], and proposed as a urinary candidate marker of ENaC activation in humans [15]. In mice, however, proteolytic ENaC activation and sodium retention in an experimental nephrotic syndrome were independent of a catalytically active CAP1/Prss8 [16]. Kidney-specific CAP1/Prss8 knockout mice maintained ENaC-mediated Na^+^ and K^+^ homeostasis, although through an aldosterone-independent mechanism when Na^+^-deprived [17]. CAP3/St14 was proposed to form direct physical interactions with CAP1/Prss8 [18], rendering its zymogen form active by proteolytic cleavage [19]. Contrarily, in human cell lines deficient for each protease, Su and coworkers demonstrated that CAP3/St14 and CAP1/Prss8 zymogen activation were not coupled [20]. The knockout models of both proteases showed similar but not identical phenotypes. It became questionable whether both proteases were part of the same proteolytic cascade, and ENaC was additionally never confirmed as a direct substrate of CAP3/St14 (see for review [7]). Concerning the proteolytic ENaC activation, treating mice with aprotinin, an unspecific serine protease inhibitor, prevented volume retention in mice with nephrotic syndrome and altered the γENaC proteolytic fragments. However, no direct inhibitory effect on channel activity was seen [21]. However, CAP3/St14 was demonstrated to be aprotinin-insensitive [5] and might thus escape inhibition.

We first generated the knockouts of nephron tubule-specific CAP3/St14 and the double knockout of CAP3/St14 and CAP1/Prss8 to study the implication of CAP3/St14 alone or together with CAP1/Prss8 in ENaC-mediated Na^+^ absorption. Next, we challenged the mice with a low Na^+^ diet, analyzed the abundance of cleaved and full-length ENaC subunits, and monitored ENaC activity following pharmacological inhibition. This should allow us to unveil further the implication of CAP1/Prss8 and CAP3/St14 in the proteolytic ENaC activation.

## 2. Materials and Methods

### 2.1. Mouse Models

Mice were kept ad libitum with food and water. At the age of 3 weeks, control and knockout mice were induced for 10 days by doxycycline (2 mg/mL doxycycline hydrochloride; Sigma, Deisenhofen, Germany) and 2% sucrose in drinking water. All animals were housed in a temperature and humidity-controlled room with an automatic 12 h light/dark cycle. If not indicated otherwise, 6–25-week-old age-matched males and females were fed on a standard (0.17% Na^+^) or low (0.01% Na^+^) Na^+^ diet (ssniff Spezialdiäten GmbH, Soest, Germany).

Animal maintenance and experimental procedures were in agreement with the Swiss federal guidelines. They were approved by the local committee for animal experimentation (Service de la Consommation et des Affaires Vétérinaires, Lausanne, Vaud, Switzerland) (#VD3775).

Inducible nephron-specific CAP3/St14 knockout mice were generated by crossing floxed CAP3/St14 (St14^flox/flox^) mice [22] with transgenic mice for Pax8-rTA^Tg/0^ and TRE-LC1^Tg/0^ [23]. The following genotypes were obtained for the control mice: St14^lox/lox^; Pax8rTA^Tg/0^; TRE-LC1^0/0^ or St14^lox/lox^; Pax8rTA^0/0^; TRE-LC1^Tg/0^ or, St14^lox/lox^; Pax8rTA^0/0^; TRE-LC1^0/0^. The following genotype was analyzed for the CAP3/St14 knockout mice (Ko): St14^lox/lox^;Pax8rTA^Tg/0^;TRE-LC1^Tg/0^. Inducible nephron-specific CAP1/Prss8 and CAP3/St14 double knockout mice (DKo) were generated by intercrossing inducible nephron-specific single CAP1/Prss8 knockout mice [17] with inducible nephron-specific CAP3/St14 mice to obtain DKo: Prss8^lox/lox^; St14^lox/lox^; Pax8rTA^Tg/0^; TRE-LC1^Tg/0^. The following genotypes were obtained for the control mice: Prss8^lox/lox^;St14^lox/lox^;Pax8rTA^Tg/0^; TRE-LC1^0/0^ or Prss8^lox/lox^; St14^lox/lox^; Pax8rTA^0/0^; TRE-LC1^Tg/0^ or, Prss8^lox/lox^; St14^lox/lox^; Pax8rTA^0/0^; TRE-LC1^0/0^. Protein lysate from inducible nephron-specific CAP1/Prss8 knockout mice was used for Western blot analysis [17].

### 2.2. Protein Extraction and Western Blot Analysis

Kidneys were collected, directly frozen in liquid nitrogen, and stored at −80 °C until protein extraction. Then, 30 μg of proteins were loaded and separated by SDS-PAGE on 10% acrylamide gels and transferred to nitrocellulose membranes (Amersham Hybond-ECL, GE Healthcare, Chicago, IL, USA). The following primary antibodies were used with dilution 1/1000: αENaC, βENaC, NCC, pNCC (T53), NHE3, NKCC2 [24,25], γENaC (StressMarq, Biosciences, Victoria, Canasa) [26,27], CAP3/St14 (R&D systems AF3946, Minneapolis, MN, USA), BK/Maxi-Ki, Kca1.1 (APC-151, Alomone labs, Jerusalem, Israel). RomK/Kir1.1 was used with the dilution of 1/10,000 for Western blots on Dko experiments and 1/1000 for Western blots on CAP3/St14 Ko experiments (kindly provided by Olivier Staub, Lausanne, Switzerland). The primary antibody for CAP1/Prss8 was used with a dilution of 1/500 (ProteinTech 15527-1-AP, Manchester, UK). Protein lysates from kidney tubular-specific α, β, or γENaC knockout mice were used as a negative control as previously described [17]. Each protein expression was controlled with their respective β-actin. Signals were revealed with ECL (Amersham ECL Western Blotting Detection Reagents, GE Healthcare, Chicago, IL, USA), and band intensity was measured using the Image Studio Lite Software (version 5.2).

### 2.3. Metabolic Cage Studies

Controls and knockout littermates received a standard diet (0.19% Na^+^) or a low salt diet (0.01% Na^+^, 10 consecutive days). Physiological parameters were measured during 4 consecutive days: Body weight (measured as % of the initial body weight at day 0 of the experiment), 24h food (g/24 h/gBW) and water intake (mL/24 h/gBW), 24 h feces (g/24 h/gBW) and urine excretion (mL/24 h/gBW), 24 h Na^+^ and K^+^ intake (mmol/gBW), and cumulative 24 h Na^+^ and K^+^ excretion (mmol). At the end of the experiment, mice were sacrificed. Organs and blood were taken. Plasma and urinary Na^+^ and K^+^ concentrations were measured with the IL943 Flame Photometer (Instrumentation Laboratory, Cheshire, UK). Plasma aldosterone concentration and renin activity were measured by the Service of Nephrology (Lausanne University Hospital (CHUV), Lausanne, Switzerland) using the Coat-A-Count RIA kit (Siemens Medical Solutions Diagnostics, Ballerup, Denmark).

### 2.4. RNAscope Analysis

Following dissection, kidneys from male C57BL/6J mice on a standard diet were fixed in 10% formalin for 24h at room temperature. RNAscope Multiplex Fluorescent V2 assay was performed according to the manufacturer’s protocol on 4 µm paraffin sections, hybridized with the probes Mm-Scnn1a-C1 (#441391, ACD, Biotechne, Minneapolis, MN, USA), Mm-Scnn1a-C3 (#441391-C3, ACD, Biotechne, Minneapolis, MN, USA), Mm-St14-C1 (#422941 ACD, Biotechne, Minneapolis, MN, USA), Mm-Prss8-O2-C3 (#59361-C3, ACD, Biotechne, Minneapolis, MN, USA), Mm 2.5 Duplex positive control polr2a and ppib (#321651, ACD, Biotechne, Minneapolis, MN, USA), 2-plex negative control DapB (#320751, ACD, Biotechne, Minneapolis, MN, USA) at 40 °C for 2 h and revealed with TSA Opal570 (#FP1488001KT, ACD, Biotechne, Minneapolis, MN, USA) or TSA Opal520 (#FP1487001KT, ACD, Biotechne, Minneapolis, MN, USA). Tissues were counterstained with DAPI and mounted with Prolong Diamond Antifade Mountant (#P36965, Thermo Fisher, Switzerland). Stainings were performed by the Histology Core Facility of the Ecole Polytechnique Fédérale de Lausanne (EPFL). For quantification, single and double positively and negatively stained cells from 20 photographic sections from 4 kidneys (*n* = 2, C57BL/6J male mice) were counterstained with DAP1 and analyzed using the ImageJ2 program.

### 2.5. Quantitative Real-Time PCR

Frozen kidney samples were homogenized using TissueLyser (Qiagen, Valencia, CA), and mRNA was isolated using the Qiagen RNeasy Mini Kit (Qiagen, Basel, Switzerland) according to the manufacturer’s instructions. The sequences of the primers used were the following: αENaC: For:5′-GCACAACCGCATGAAGACG-3′, Rev:5′-AAAGCAAACTGCCAGTACATC-3′ (Primer 3 Plus); βENaC: For:5′-ACCCGGTGGTTCTCAATTTG-3′, Rev:5′-ACACAGTTCCATTGGCACTG-3′ (Primer 3 Plus); γENaC: For:5′-CCGAGATCGAGACAGCAATGT-3′, Rev:5′-CGCTCAGCTTGAAGGATTCTG-3′ (Primer 3 Plus); NCC: For:5′-ACACGGCAGCACCTTATACAT-3′, Rev:5′-GAGGAATGAATGCAGGTCAGC-3′ (PB-ID: 14547897a1); NKCC2: For:5′-GTCTCGGTGTGATTATCATCGG-3′, Rev:5′-ATCCGTTTGTGGCGATAGCAG-3′ (PB-ID: 1079519a1); HE4: For:5′-AGGTCAAGTCTCCACGAAGCCA-3′, Rev:5′-AGAACACTGGCTGTCCACCTGA-3′ (ORiGENE MP218431); PAI-1: For:5′-CCTCTTCCACAAGTCTGATGGC-3′, Rev:5′-GCAGTTCCACAACGTCATACTCG-3′ (ORiGENE MP215217); HAI-1: For:5′-CTTCGTGAGGAAGAGTGCATGC-3′, Rev:5′-TCACACTCCAGGAAGCCATCGA-3′ (OriGENE MP215250); HAI-2: For:5′-ATGGAGGCTGTGAAGGCAATGG-3′, Rev:5′-GGACAGAAGAGTCGGCTCCATT-3′ (ORiGENE MP215251); GAPDH: For:5′-AGGTCGGTGTGAACGGATTTG-3′, Rev:5′-TGTAGACCATGTAGTTGAGGTCA-3′ (PB-ID: 6679937a1). Real-time PCR was performed using Power SYBRgreen PCR Master Mix (Applied Biosystems, Thermo Fisher Scientific, Warrington, UK) and run on a 7500 Fast Applied Biosystems (Applied Biosystems, Thermo Fisher Scientific, Warrington, UK). Each measurement was performed in triplicate and normalized to GAPDH.

### 2.6. Diuretic Treatments

Before treatment, control and CAP3/St14 Ko and CAP1/Prss8; CAP3/St14 DKo mice were subjected for 10 consecutive days to a low salt diet (0.01% Na^+^). Then, mice received a single intraperitoneal injection of either benzamil (0.2 ug/gBW, B2417, Sigma-Aldrich, Deisenhofen, Germany), furosemide (20 mg/kgBW, F4381, Sigma-Aldrich, Deisenhofen, Germany), hydrochlorothiazide (20 mg/kgBW, H4754, Sigma-Aldrich, Deisenhofen, Germany), or vehicle (0.5% DMSO + 0.9% saline). Urine was collected during the 6 h post-injection. Blood and organs were harvested at the end of the experiment. Urinary and plasma Na^+^ and creatinine concentrations were quantified to determine the fractional Na^+^ excretion FE(Na^+^) as previously described [17].

### 2.7. Statistical Analyses

GraphPad Prism was used to perform all statistical analyses and data are presented as scatter plots. A repeated measure two-way ANOVA with post hoc Šìdák multiple comparison test was performed to evaluate the changes in the body weight of mice, 24 h Na^+^ and K^+^ intake, and the cumulative 24 h Na^+^ and K^+^ excretion. Plasma electrolytes, aldosterone concentrations, protein expression, and RNAscope data were analyzed using an unpaired two-tailed Welch’s *t*-test. Results are represented as mean ± SEM, and *p*-value < 0.05 was considered statistically significant; * *p* < 0.05; ** *p* < 0.01, *** *p* < 0.001.

## 3. Results

### 3.1. CAP3/St14 Tubule Specific Deficiency Changed Protein Abundances of ENaC Subunits but Did Not Impair Na^+^ Homeostasis

Deletion of the serine protease CAP3/St14 in the nephron tubules resulted in a significant reduction of CAP3 protein abundance in the whole kidney (Figure 1A,B). Under Na^+^ deprivation, CAP3/St14 knockout mice showed several changes in the abundances of Na^+^ transporting proteins. αENaC, cleaved αENaC, and NKCC2 protein expression were upregulated in these mice (Figure 1A,B). Full-length βENaC and γENaC, as well as pNCC, were significantly lowered (Figure 1A,B), although mRNA transcript expression of α, β, and γENaC subunits did not vary (Appendix A). Interestingly, upon Na^+^ deprivation, CAP1/Prss8 protein abundance was also significantly downregulated (Figure 1A,B). CAP3/St14 was significantly upregulated in CAP1/Prss8 (CAP1/Prss8^Pax8LC1^) knockout mice on a standard Na^+^ diet but not a low Na^+^ diet (Appendix A)**.** This suggested that at least on the protein level, CAP1/Prss8 is inhibiting CAP3/St14, while CAP3/St14 is stimulating CAP1/Prss8 protein abundance.

Protein expression of βENaC and the cleaved γENaC were significantly down-, and the phosphorylated form of the sodium chloride cotransporter (pNCC) and NKCC2 both upregulated in CAP3/St14 Ko mice on standard diet (Appendix A).

To study whether the renal CAP3/St14 was implicated in ENaC-mediated Na^+^ absorption, we subjected CAP3/St14 Ko (CAP3/St14^Pax8LC1^) and their littermate controls to a low Na^+^ diet and measured renal physiological parameters. Overall, CAP3/St14 Ko mice maintained Na^+^ and K^+^ balance, although there is a tendency to decrease cumulative Na^+^ excretion and plasma Na^+^ concentration. No statistical difference was observed in the CAP3/St14 Ko compared to control mice in the 24 h Na^+^ and K^+^ intake, the cumulative 24 h Na^+^ and K^+^ excretion, as well as the plasma Na^+^, K^+^, and aldosterone levels and renin activity when Na^+^-deprived (Figure 2). Metabolic parameters such as 24 h food and water intake, 24 h feces, urine excretion, or 24 h urinary Na^+^ and K^+^ excretion did not differ (Appendix A). This confirmed data obtained from CAP3/St14 Ko and control mice on a standard diet (Appendix A).

### 3.2. ENaC Is Highly Co-Expressed with CAP3/St14 and Less with CAP1/Prss8 in Distal Tubules

We next investigated whether CAP3/St14 and CAP1/Prss8 were expressed in the same distal tubules as ENaC. Using the RNAscope-based technology, we examined the spatial localization of CAP3 (St14), CAP1 (Prss8), and αENaC (Scnn1a) transcript expression in kidney cortex from wild-type mice on a standard diet (Figure 3). mRNA transcript expression of St14 was exclusively found in Scnn1a positive distal tubules cells, thereby highly overlapping (Figure 3A,D), whereas Prss8 mRNA transcripts were predominantly enriched in the proximal tubules, but also found together with Scnn1a transcripts although more sparsely (Figure 3B,E). St14 is only co-expressed with Scnn1a in the distal tubules (Figure 3C,F). Positive (POLR2A and PPIB) and negative controls (Dapb) are represented in Appendix A.

### 3.3. CAP1/Prss8; CAP3/St14 DKo Mice Restored ENaC Subunit Protein Abundances and Aldosterone Regulation of ENaC but Not of NCC

Knockout of CAP3/St14 and CAP1/Prss8 in renal tubules resulted in the near deletion of both proteins (Figure 4A,B). Under a low Na^+^ diet, the abundances of several Na^+^ and K^+^ transporting proteins were unchanged, except for pNCC and BK, which were significantly reduced in DKo (Figure 4A,B). Protein abundances, as well as mRNA transcript expression of ENaC subunits, NCC, and NKCC2, were not different from the controls (Figure 4 and Appendix A).

To further evaluate the consequences of CAP1/Prss8 and CAP3/St14 deficiencies in the renal tubules, adult induced DKo mice were fed with a low Na^+^ diet. Data of the physiological parameters clearly revealed that both DKo and control mice are not different with respect to body weight changes, 24 h Na^+^ and K^+^ intake, cumulative 24 h Na^+^ and K^+^ excretion, as well as plasma Na^+^, K^+^, and surprisingly, also the aldosterone levels and renin activity (Figure 5). These results thus differed from the previously observed phenotype of the single CAP1/Prss8^Pax8LC1^ knockout [17]. Metabolic parameters such as 24 h food and water intake, 24 h feces excretion and urine volume, and 24 h urinary Na^+^ and K^+^ excretion did not differ between controls and DKo mice (Appendix A).

Under a standard diet, CAP1/Prss8; CAP3/St14 double knockout mice restored increased pNCC and decreased BK protein expression, as observed under a low salt diet (Appendix A). Neither metabolic nor physiological parameters from CAP1/Prss8; CAP3/St14 DKo mice differed from their control littermates (Appendix A).

To see whether the alteration of protein expression in ENaC subunits as well as pNCC and NKCC2 in CAP3/St14 Ko and CAP1/Prss8; CAP3/St14 DKo mice could have an impact on their activity, diuretics were administrated following the consecutive 10 days of Na^+^-deprived diet. The natriuretic response to acute ENaC and NKCC2 inhibition by benzamil and furosemide, respectively, did not change between CAP3/St14 Ko mice, CAP1/Prss8; CAP3/St14 DKo mice and control littermates (Figure 6A–D). Interestingly, the inhibition of NCC by thiazide showed significantly decreased activity in CAP1/Prss8; CAP3/St14 DKo mice compared to single CAP3/St14 Ko (Figure 6E,F) and thus correlated with decreased protein expression in CAP1/Prss8; CAP3/St14 DKo mice (Figure 4).

To summarize, the CAP1/Prss8; CAP3/St14 DKo mice restored the aldosterone-induced ENaC regulation that was uncoupled in CAP1/Prss8 deficient mice and, except for pNCC, normalized the protein abundances of altered renal Na^+^ and K^+^ transporting proteins in single CAP3/St14 knockout mice independent of the diet conditions. Deficiency of both proteases significantly affected the NCC activity while maintaining Na^+^ balance upon Na^+^ deprivation.

## 4. Discussion

### 4.1. CAP3/St14 Was Not Required for Proteolytic ENaC Activation, Albeit Affecting Its Protein Abundance

Identified as in vitro ENaC channel activating proteases, the membrane-bound serine proteases CAP1/Prss8 or CAP3/St14 were amongst the first shown to induce a significant increase of Na^+^ current when co-expressed with rat ENaC subunits in Xenopus oocytes [6]. In this in vitro system, ENaC activation was dependent on the catalytically active CAP3, which mediated cleavage of ENaC at basic residues near the γENaC furin site [28]. The in vivo regulation of ENaC by CAP3/St14 was never demonstrated. Deficiency of CAP3/St14 within the kidney tubules neither changed the cleavage pattern (Figure 1) nor affected Na^+^ and/or K^+^ balance (Figure 2). Despite similarly increased plasma aldosterone levels in CAP3/St14 Ko and controls under Na^+^ deprivation (Figure 2), α, β, and γENaC abundances were significantly altered in CAP3/St14 Ko mice (Figure 1).

While renal CAP1/Prss8 is significantly downregulated in CAP3/St14 Ko mice on a low Na^+^ diet (Figure 1), CAP3/St14 protein abundance is significantly upregulated in CAP1/Prss8^Pax8LC1^ knockout mice (Appendix A) indicating that renal CAP1/Prss8 inhibited CAP3/St14, while renal CAP3/St14 promoted CAP1/Prss8 protein expression. Thereby, ENaC activity seemed to be equally induced in CAP3/St14 knockout and controls, as evidenced by similar urinary Na^+^ retention and similarly increased plasma aldosterone levels (Figure 2). The protein expression levels of all three ENaC subunits were altered diet-independently without affecting the mRNA transcript expression (Figure 1 and Appendix A). Interestingly, the Na^+^-Cl^−^ cotransporter seemed less active, as evidenced by reduced pNCC protein abundance, indicating that a common pathway of ENaC and NCC stimulation might be affected as ENaC activity modulated NCC in cirrhotic mice [29]. The increased Na^+^-K^+^-Cl^−^ cotransporter protein abundance might compensate for the reduced NCC activity (Figure 1).

We did not observe changes in the cleavage pattern of ENaC subunits in mice deficient for CAP3/St14 or the CAP1/Prss8; CAP3/St14 double knockout mice compared to their corresponding controls (Figure 1 and Figure 4) and confirmed previous results performed in kidney-specific CAP1/Prss8 knockout mice [17]. Apart from an apparent inhibition of CAP3/St14 by CAP1/Prss8 and stimulation of CAP1/Prss8 by CAP3/St14 in whole kidney lysates, we found no in vivo evidence for a common proteolytic cascade. CAP1/Prss8; CAP3/St14 double knockout mice restored the aldosterone-dependence of ENaC that was uncoupled in CAP1/Prss8 Ko mice [17]. In kidneys from wild-type mice on a standard diet, the protease CAP1/Prss8 showed a higher mRNA transcript expression in proximal and distal renal tubules, whereas CAP3/St14 mRNA transcript expression was restricted to the distal tubules. Still, both proteases are co-expressed with αENaC (Figure 3).

We instead propose that either protease affected a likely common substrate along the ENaC activating pathway that is not yet identified. Interestingly, the mRNA transcript expression of the serine protease inhibitors HE4, PAI-1, and HAI-1 but not HAI-2 was significantly upregulated in CAP3/St14 but unchanged in double CAP3/St14; CAP1/Prss8 knockout mice (Appendix A). The role of these inhibitors in the observed differences in protein abundances is still unknown and will require further investigation. In this context, it is worth mentioning that in CAP1/Prss8 as well CAP3/St14 knockout mouse models, several tight junction proteins were found altered in the corresponding epithelia (see for review [7]), and cell adhesion molecules like EpCAM [30] might present a common target.

### 4.2. Aldosterone-Dependent ENaC Activation, but Not NCC Activity Was Restored in Na^+^-Deprived CAP1/Prss8;CAP3/St14 Dko Mice

Serine proteases play diverse roles in regulating biological processes ranging from embryonic development to impaired epithelial barrier function to cancer metastasis, likely with differential spatial and timely restricted expression (see for review [7]). Surprisingly, double knockout mice for CAP1/Prss8 and CAP3/St14 no longer showed disturbed protein abundances, as seen in the single CAP3/St14 knockout mice, in addition to increased renal CAP1/Prss8 protein abundance (Figure 1 and Figure 4). Both overexpression and deletion of CAP1/Prss8 in the mouse epidermis led to impaired epithelial barrier function and severe dehydration (see for review [7]), which might indicate the required balance of protease expression. Deletion of CAP3/St14 resulted in a similar, although not identical, phenotype, whereas epidermal overexpression of CAP3/St14 resulted in multistage carcinogenesis that could be completely negated by overexpression of its endogenous inhibitor HAI-1 (see for review [7]). This was never observed in CAP1/Prss8 overexpressing mice [31]. It was, however, not reported whether CAP3/St14 and CAP1/Prss8 protein abundancies were upregulated in CAP1/Prss8 versus CAP3/St14 mutant mice. Hypomorphic CAP1/Prss8 (frizzy; fr/fr) mice restored normal development of HAI-1 (Spint1)-deficient mice [32]. Conversely, mutations in human SPINT2 were associated with congenital tufting enteropathy characterized by severe intestinal dysfunction with induced EpCAM cleavage and decreased claudin-7 expression that resulted in organoid rupture [33]. The clinical features could be prevented by intestinal-specific inactivation of CAP3/St14 [34]. Overall, this indicated that tight regulation of protease and protease inhibitors might be required, which might be independent of the same proteolytic pathway. In the same line, corrected phenotypes were previously reported through additional gene targeting of a protein from the same family, e.g., rescue of hypomagnesemia in claudin-10/claudin-16 double knockout mice [35], highlighting the complexity of claudin interactions. Furthermore, mice with a cardiomyocyte-specific knockout for the glucocorticoid and mineralocorticoid receptor signaling exhibited improved life span compared to single knockout mice [36].

Our data suggest that CAP1/Prss8 and CAP3/St14 protein abundancies are linked without being part of the same proteolytic pathway. ENaC activity is conserved, as evidenced by the ENaC-specific diuretic treatment of benzamil. Both proteases are not essential for proteolytic ENaC activation, but CAP3/St14 is required to maintain NCC activity following Na^+^ deprivation. Therefore, the presented study will help decipher the complexity of these channel-activating protease cascades and their role in controlling ENaC- and NCC-mediated Na^+^ absorption.

## Figures and Tables

**Figure 1 cells-12-02342-f001:**
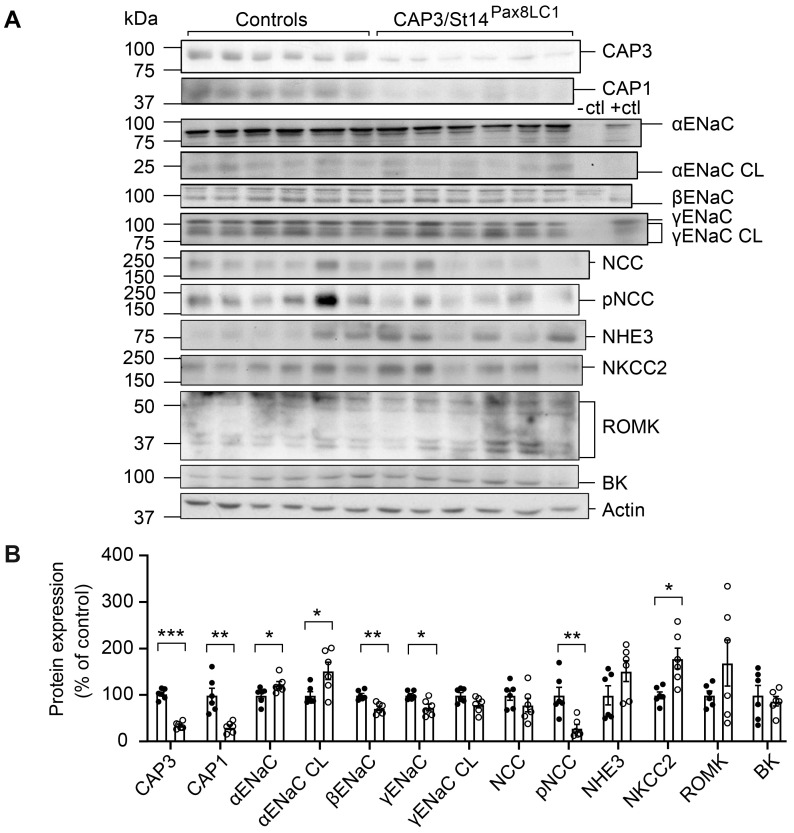
CAP3/St14 knockout mice showed altered protein abundancies for ENaC subunits, NCC, and NKCC2 and downregulation of CAP1/Prss8 upon Na^+^ deprivation. (**A**) Representative Western blot analyses of CAP3, CAP1, αENaC, αENaC CL (cleaved), βENaC, γENaC, γENaC CL (cleaved), NCC, pNCC, NHE3, NKCC2, ROMK and BK on kidney lysates from controls (black circles, *n* = 6) and CAP3/St14 knockout (CAP3/St14^Pax8LC1^, white circles, *n* = 6). Kidney lysates of control and renal tubular-specific knockouts of αENaC, βENaC, and γENaC served as negative (-ctl) and positive (+ctl) controls, respectively. (**B**) Quantification of data. Results are presented as mean ± SEM. Data were analyzed using an unpaired two-tailed Welch’s *t*-test, and *p* values <0.05 were considered statistically significant; * *p* < 0.05, ** *p* < 0.01, *** *p* < 0.001.

**Figure 2 cells-12-02342-f002:**
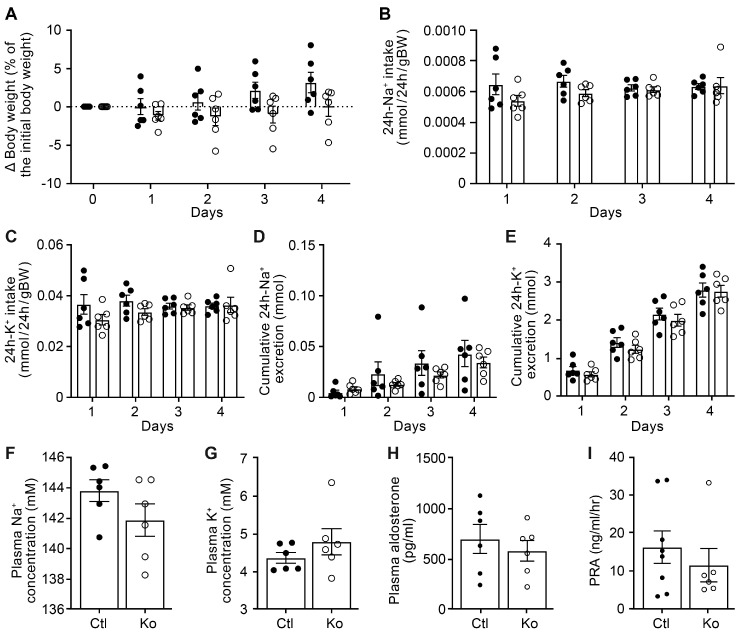
CAP3/St14 knockout mice displayed normal Na^+^ and K^+^ handling under a low Na^+^ diet. (**A**) Body weight changes (expressed as a percentage of initial body weight). (**B**) 24 h Na^+^ and (**C**) K^+^ intake (mmol/24 h/gBW). (**D**) 24 h cumulative Na^+^ and (**E**) K^+^ excretion (mmol). (**F**) Plasma Na^+^ and (**G**), K^+^ (mM), (**H**) aldosterone concentration (pg/mL), and (**I**) renin activity (ng/mL/h) in control (Ctl, *n* = 6−8, black circle) and CAP3/St14 knockout (Ko, *n* = 6, white circles) mice. Results are presented as mean ± SEM. (**A**–**E**) were analyzed by a two-way ANOVA with a post hoc Šìdák multiple comparison test. (**F**–**I**) were analyzed by an unpaired two-tailed Welch’s *t*-test.

**Figure 3 cells-12-02342-f003:**
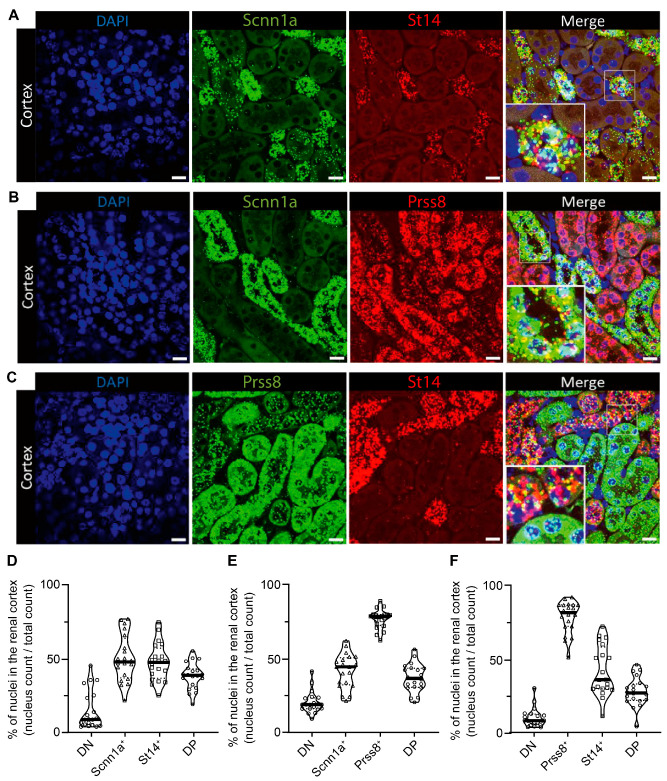
RNAscope-based co-expression of St14 (CAP3), Prss8 (CAP1), and Scnn1a (αENaC) mRNA transcripts in the renal cortex of wild-type mice under a standard diet. (**A**) Visualization of nuclei (DAPI staining, left) and expression of Scnn1a (green, middle left) and St14 (red fluorescent channel, middle right), and both (merged picture, right), (**B**) counterstaining with DAPI (left), detection of Scnn1a (green, middle left) and Prss8 (red fluorescent channel, middle right), and both together (merged picture, right), and (**C**) staining with DAPI (left), Prss8 (green, middle left), St14, (red fluorescent channel, middle right), and the merged pictures (right). (**D**–**F**) Corresponding quantification of double negative (DN) and positive (DP) as well as single (**D**) Scnn1a^+^ and St14^+^, (**E**) single αENaC^+^ and Prss8^+^ and (**F**) single Prss8^+^ and St14^+^ positive cells expressed as a percentage of positively stained divided by the total number of nuclei and illustrated as a violin blot. Magnification: 40×. The scale bar represents 20 µm. Data were analyzed with an unpaired two-tailed Welch’s *t*-test.

**Figure 4 cells-12-02342-f004:**
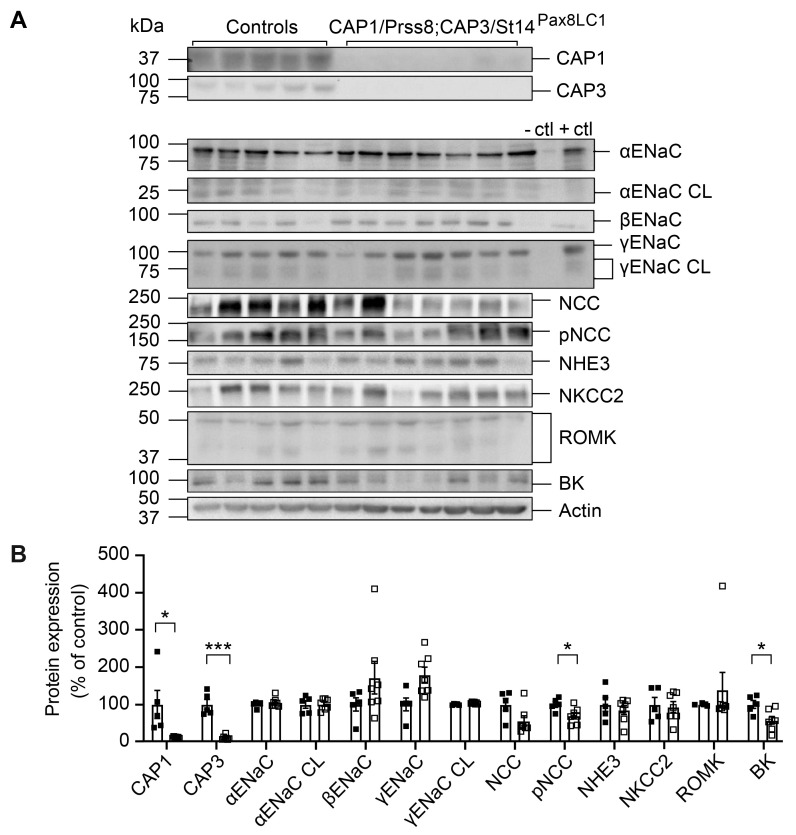
CAP1/Prss8; CAP3/St14 double knockout mice showed normalized ENaC subunit but still lowered pNCC and BK protein abundances under a low Na^+^ diet. (**A**) Representative Western blot analysis of CAP3, CAP1, αENaC, αENaC CL (cleaved), βENaC, γENaC, γENaC CL (cleaved), NCC, pNCC, NHE3, NKCC2, ROMK, and BK on kidney lysates from controls (black squares, *n* = 5) and CAP1/Prss8; CAP3/St14 double knockout (CAP1/Prss8; CAP3/St14^Pax8LC1^, *n* = 7) mice. Kidney lysates of control and renal tubular-specific knockouts of αENaC, βENaC, and γENaC served as negative (−ctl) and positive (+ctl) controls, respectively. (**B**) Quantification of the data. Results are presented as mean ± SEM. Data were analyzed using an unpaired two-tailed Welch’s *t*-test. *p* values < 0.05 were considered statistically significant; * *p* < 0.05, *** *p* < 0.001.

**Figure 5 cells-12-02342-f005:**
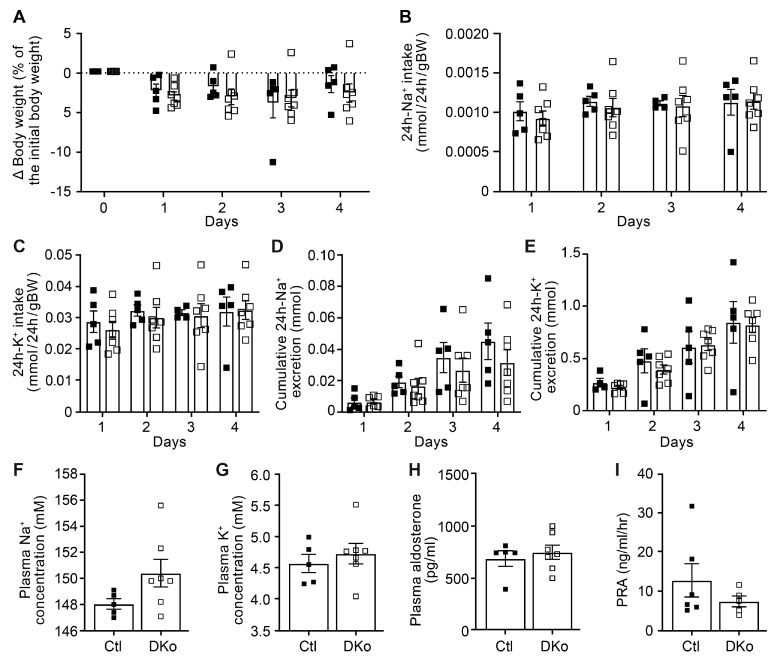
CAP1/Prss8; CAP3/St14 double knockout mice displayed normal Na^+^ and K^+^ handling under a low Na^+^ diet. (**A**) Body weight changes (expressed as a percentage of initial body weight). (**B**) 24 h Na^+^ and (**C**) K^+^ intake (mmol/24 h/gBW). (**D**) 24 h cumulative Na^+^ and (**E**) K^+^ excretion (mmol). (**F**) Plasma Na^+^ and (**G**) K^+^ (mM), (**H**) aldosterone concentration (pg/mL), and (**I**) renin activity (ng/mL/h) in control (Ctl, black square, *n* = 5–6) and CAP1/Prss8; CAP3/St14 double knockout mice (DKo, white square, *n* = 5–7). Results are presented as mean ± SEM. (**A**–**E**) were analyzed by a two-way ANOVA with a post hoc Šìdák multiple comparison test. (**F**–**I**) were analyzed by an unpaired two-tailed Welch’s *t*-test.

**Figure 6 cells-12-02342-f006:**
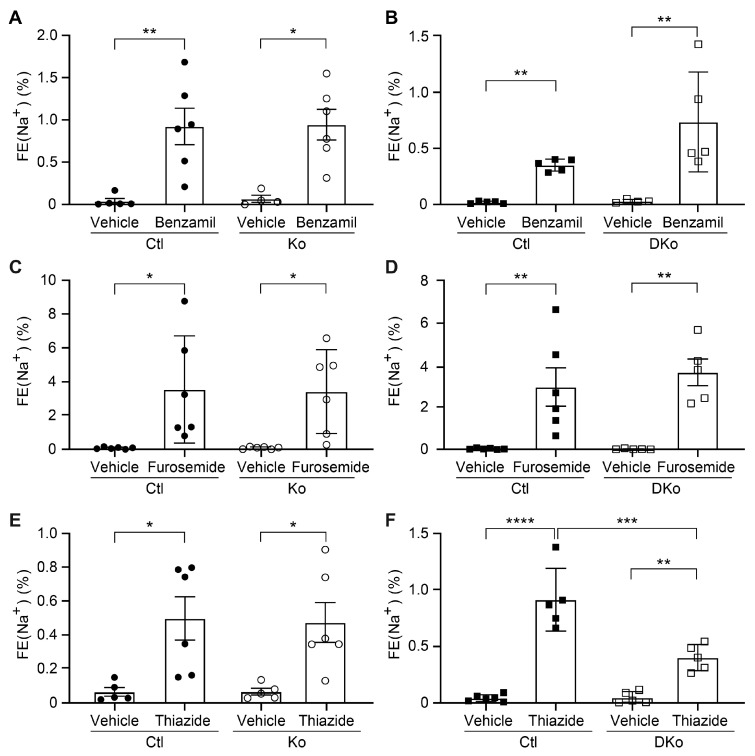
NCC activity was decreased in CAP1/Prss8; CAP3/St14 double knockout mice under a low Na^+^ diet. Natriuretic response expressed as fractional excretion of Na^+^ (in %) in CAP3/St14 control (Ctl, black circle, *n* = 5–6), CAP3/St14 knockout (Ko, white circle, *n* = 4–6), CAP1/Prss8; CAP3/St14 control (Ctl, black square, *n* = 5–6) and CAP1/Prss8; CAP3/St14 double knockout (Dko, white square, *n* = 5–6) mice. Mice were treated with vehicle or (**A**,**B**) benzamil (0.2 μg/gBW), (**C**,**D**) furosemide (20mg/kgBW), or (**E**,**F**) hydrochlorothiazide (20 mg/kgBW). Results are presented as mean ± SEM. Data were analyzed using a two-way ANOVA with Sidak’s multiple comparison test (column factor: genotype; row factor: treatment). *p* values < 0.05 were considered statistically significant. * *p* < 0.05, ** *p* < 0.01, *** *p* < 0.001, **** *p* < 0.0001.

## Data Availability

Data are available on request.

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
