# Peer review of "Kidney-Specific Membrane-Bound Serine Proteases CAP1/Prss8 and CAP3/St14 Affect ENaC Subunit Abundances but Not Its Activity"

_cells, 2023, doi:10.3390/cells12192342_

Round 1

Reviewer 1 Report

In this manuscript, Elodie Ehret and his colleagues reveal a complex regulation of ENaC by these serine proteases on the expression level rather than on its proteolytic activation.This study is interesting, innovative, and of good clinical relevance and could be published in the cells

Author Response

Responses to Editor and Referees

We would like to thank the reviewers for their encouraging and constructive comments and suggestions. We have now included the following modifications:

  • All “data not shown” are now added as Supplementary figures S1, S2, S4, S6, S8, S10.
  • New data about the mRNA transcript expression of serine protease inhibitors were generated and are included as new Supplementary figure S12.
  • Figures and data were arranged and discussed as suggested.
  • Text was amended as recommended by the reviewer.

Comments from Reviewer #1

In this manuscript, Elodie Ehret and his colleagues reveal a complex regulation of ENaC by these serine proteases on the expression level rather than on its proteolytic activation. This study is interesting, innovative, and of good clinical relevance and could be published in the cells.

We thank the reviewer for this positive comment.

Reviewer 2 Report

Title: Kidney-specific membrane-bound serine proteases CAP1/Prss8 and CAP3/St14 affect ENaC subunit abundances but not its activity

Authors: Ehret E., et al.

This manuscript investigates kidney-specific knockouts of CAP1 and CAP3 in order to determine their respective contribution to regulating ENaC activity and thereby sodium resorption.

The Authors should stick to the use of one nomenclature throughout the manuscript. It is unfortunate that both proteins have three names, but deciding for one would make it easier for the reader to follow.

The last paragraph of the Introduction (lines 77-86) should be rewritten. The sentences are quite long and the text is difficult to follow. The Authors allude to results before the data have been presented. Perhaps it would be better to end with the question that the Author’s intend to investigate.

I think it essential that the Authors first demonstrate the respective knockouts for the genes of interest in their mouse models at both the mRNA and protein level before presenting functional data on these mice. Some of the data are contained with the supplement, but are not referred to in the text in this context.

When describing the initial treatment of the CAP3 knockout mice, normal and low sodium diets, the Authors’ state twice “(data not shown)” lines 185 and 186. Why? What are supplemental data files for?

Line 204 The Authors state, “CAP3 was significantly upregulated in CAP1 Ko (Prss8PaxLC1) knockout mice…” However, these mice are not presented in Figure 2A. If these are the unpublished data to which the Authors refer, then why not include them in the supplementary figures? As the data are presented now, it is impossible to evaluate the Authors statement on CAP1 and CAP3 expression.

Figure 3 the Authors investigate mRNA expression levels of St14 and Prss8 to argue for co-expression. In my experience, mRNA does not always correlate with protein expression. Why not use immune fluorescence?

Figure 4 Again, perhaps it is a matter of style, but I prefer to see that knockout established before evaluating the functional data.

Have the Authors considered that the differences they are looking for might not be on the level of the serine proteases, but rather at the level of serine protease inhibitors, e.g. HE4 or PAI-1?

Minor points

Line 59 define HAI-1 and HAI-2

Line 85 define NCC

Figure S2 for the αENaC blot, there is no band visible in the positive control (+ctl). Why should I trust the specificity of this antibody? Furthermore, in the original image files most αENaC blots show a single band. Why does this blot show a doublet? Also, the MW are not indicated on any of the original images, nor the band of interest, making it extremely difficult to evaluate the data.

Line 258 define BK. The Authors have an abundance of undefined abbreviations, please consider including a list of abbreviations, as this might make things easier for the reader.

Author Response

Responses to Editor and Referees

We would like to thank the reviewers for their encouraging and constructive comments and suggestions. We have now included the following modifications:

  • All “data not shown” are now added as Supplementary figures S1, S2, S4, S6, S8, S10.
  • New data about the mRNA transcript expression of serine protease inhibitors were generated and are included as new Supplementary figure S12.
  • Figures and data were arranged and discussed as suggested.
  • Text was amended as recommended by the reviewer.

Comments for Reviewer #2

This manuscript investigates kidney-specific knockouts of CAP1 and CAP3 in order to determine their respective contribution to regulating ENaC activity and thereby sodium resorption.

The Authors should stick to the use of one nomenclature throughout the manuscript. It is unfortunate that both proteins have three names, but deciding for one would make it easier for the reader to follow.

We have now standardized the nomenclature throughout the manuscript.

The last paragraph of the Introduction (lines 77-86) should be rewritten. The sentences are quite long and the text is difficult to follow. The Authors allude to results before the data have been presented. Perhaps it would be better to end with the question that the Author’s intend to investigate.

This paragraph is now rewritten.

I think it essential that the Authors first demonstrate the respective knockouts for the genes of interest in their mouse models at both the mRNA and protein level before presenting functional data on these mice. Some of the data are contained with the supplement, but are not referred to in the text in this context.

We now reorganized the figures as proposed. All the figures are now referred within the text.

When describing the initial treatment of the CAP3 knockout mice, normal and low sodium diets, the Authors’ state twice “(data not shown)” lines 185 and 186. Why? What are supplemental data files for?

These data are now shown as new supplementary figures S4 and S6.

Line 204 The Authors state, “CAP3 was significantly upregulated in CAP1 Ko (Prss8PaxLC1) knockout mice…” However, these mice are not presented in Figure 2A. If these are the unpublished data to which the Authors refer, then why not include them in the supplementary figures? As the data are presented now, it is impossible to evaluate the Authors statement on CAP1 and CAP3 expression.

The data are now added as supplementary figure S2.

Figure 3 the Authors investigate mRNA expression levels of St14 and Prss8 to argue for co-expression. In my experience, mRNA does not always correlate with protein expression. Why not use immune fluorescence?

Commercial and homemade antibodies unfortunately did not work. mRNA transcript expression may not always correlate with protein expression although it gave us important insights in the spatial distribution with the kidney tubules. We were specifically interested in CAP1/Prss8 and CAP3/St14 expression with (a)ENaC. This is now better specified in the manuscript.

Figure 4 Again, perhaps it is a matter of style, but I prefer to see that knockout established before evaluating the functional data.

As suggested by the reviewer, we switched the concerned figures and amended the text accordingly.

Have the Authors considered that the differences they are looking for might not be on the level of the serine proteases, but rather at the level of serine protease inhibitors, e.g. HE4 or PAI-1?

We newly performed qPCR for the serine protease inhibitors human epididymis protein-4 (HE4) and the plasminogen activator inhibitor-1 (PAI-1) as suggested and added the hepatocyte growth factor activator inhibitor type 1 (HAI-1) and type 2 (HAI-2). Interestingly, HE4, PAI-1, HAI-1 are significantly upregulated at the mRNA transcript expression level in kidneys of CAP3/St14 knockout mice. These data are now included as Supplementary Figure 12 and discussed.

Minor points

Line 59 define HAI-1 and HAI-2

This has now specified. Additionally, a list of abbreviations was added in materials & methods.

Line 85 define NCC

This has now been defined. A list of abbreviations is now added.

Figure S2 for the αENaC blot, there is no band visible in the positive control (+ctl). Why should I trust the specificity of this antibody? Furthermore, in the original image files most αENaC blots show a single band. Why does this blot show a doublet? Also, the MW are not indicated on any of the original images, nor the band of interest, making it extremely difficult to evaluate the data.

We repeated the blot for the alpha ENaC antibody and the positive control showed up. The alpha ENaC antibody was first described by Sorensen and Loffing, 2013, Kidney International), used in several publications, and was recently reconfirmed (Mutchler et al., 2023, Physiological Reports). We always used protein lysates from positive (+/+) and negative (-/-) controls to determine the full length band. The molecular weight are now indicated on all original images.

Line 258 define BK. The Authors have an abundance of undefined abbreviations, please consider including a list of abbreviations, as this might make things easier for the reader.

A list of abbreviations is now added.

Round 2

Reviewer 2 Report

The Authors have addressed my concerns.